# Using Social Network Analysis to Assess ‘Groupness’ in a Mixed-Species Zoo Exhibit of Tufted Capuchins (*Sapajus apella*) and Squirrel Monkeys (*Saimiri sciureus*)

**DOI:** 10.3390/ani14233360

**Published:** 2024-11-22

**Authors:** Sophia Daoudi-Simison, Phyllis Lee, Hannah M. Buchanan-Smith

**Affiliations:** 1School of Psychology, Faculty of Medical Sciences, Newcastle University, Newcastle upon Tyne NE2 4DR, UK; 2Psychology, Faculty of Natural Sciences, University of Stirling, Stirling FK9 4LA, UK; phyllis.lee@stir.ac.uk (P.L.); h.m.buchanan-smith@stir.ac.uk (H.M.B.-S.)

**Keywords:** mixed-species groups, primates, capuchins, squirrel monkeys, social network analysis

## Abstract

This study investigated the associations between two different types of monkeys—tufted capuchins and squirrel monkeys—living together in captivity. Understanding how these mixed-species groups behave is crucial for informing effective husbandry practices that can ultimately enhance welfare practices. We employed social network analysis, a quantitative method for measuring relationships and behaviours within groups, to observe these monkeys at the Royal Zoological Society of Scotland’s Living Links to Human Evolution Research Centre, Edinburgh Zoo. Over 183 h, we monitored how frequently the monkeys came into proximity with one another. Our findings indicate that while the two species shared a space, they did not function as a cohesive social group, as both groups exhibited low rates of mixed-species associations. These results suggest that simply sharing an environment is insufficient for the animals to be considered a group. By better understanding the ecological pressures that favour mixed-species groups in the wild, we can apply this knowledge in captivity to encourage more naturalistic behaviours.

## 1. Introduction

There are a number of taxa that have been reported to form mixed-species groups in the wild, particularly birds, fish and mammals [1]. The ‘dual benefit’ of grouping (i.e., reduced risk of predation and increased foraging success) has been provided as an explanation for this natural phenomenon [2]. However, it is often unclear how to determine what makes a group, especially when considering mixed-species. In the existing literature, there appears to be agreement on the following prerequisites for animal ‘grouping’ [3,4,5], which can be applied to both single-species and mixed-species groups, namely, there should be: (1) group stability, (2) an element of socialness [6] that is, interactions, (3) proximity and behavioural coordination in time and space, and (4) a minimum of two individuals. The first three require elaboration on how to define and measure whether these prerequisites are met. Group stability refers to the group maintaining a recognisable configuration over time, even as individual members join or leave. While the relative positions of individuals may shift, a degree of continuity among members is essential. The element of socialness may be directly observed by social attraction, where individuals seek out the presence of others in the group, which also serves to maintain the group. This association may be direct, such as physical contact or social behaviours, such as play and grooming, or indirect, such as through vocal communication, with physical proximity serving as a useful proxy for social engagement. Lastly, behavioural coordination in time and space implies that individuals are engaged in similar activities simultaneously and within a meaningful distance, allowing for both direct (e.g., physical) and indirect (e.g., vocal or visual) interactions. However, the ability to quantify social structure and the mechanisms and functions that underpin it has long been a challenge within the area of ethology, especially when considering the link between individual and group behaviour [7,8,9].

Patterns of association can arise due to individuals sharing identical or similar motivations and behaviours, such as the degree to which they are hungry, thirsty or how they respond to a potential threat [7,8,9,10]. Yet we know that social relationships and social structure are more complex than this, with individuals often engaging in non-random interactions and movements that require the development of behavioural strategies, resulting in the formation of a multi-layered and multi-behavioural structure [10,11]. Socially complex species such as non-human primates have a diverse repertoire of social behaviours and interactions that differ in type (e.g., cooperative, antagonistic, or sexual) as well as frequency and duration (e.g., years, days or minutes). Furthermore, the type of interactions that occur will likely depend on factors such as kinship, dominance, body size, sex, age, and general health (e.g., parasite load) of the participating individuals [7,8,10]. In the absence of kinship (e.g., unrelated conspecifics or heterospecifics), social behaviour can be explained by mutualism (i.e., shared benefits) and manipulation (i.e., benefits are disproportionate [12,13]). Therefore, to better understand social structure and the link between the behaviour of individuals within mixed-species groups, we need to analyse the relationships of all group members [7,11,14,15].

While this is not a new concept, within-group, between-individual links are often difficult to measure quantitatively. One approach is the use of social network analysis, based on mathematical graph theory [7,8,9], that examines social relationships (direct and indirect) via the construction of networks. Using social network analysis to gain insights into the position and role of the individual within a group enables building a more complete picture of the complex sociality and structure of group-living animals. In a captive setting, it has been used to examine the impact of environmental change, such as moving to a new enclosure [16], spatial association and social preferences [17,18], demographic changes (births, deaths, maturation [19]) and how manipulating group composition can improve social stability and reduce levels of intra-specific aggression [20,21].

Thus far, social network analysis has been applied to the study of mixed-species bird flocks [15,22,23,24], fish assemblages affected by non-native species invasions [25], and mixed-species groups of savannah herbivores [26]. Of the platyrrhine primates to form mixed-species groups, the most commonly reported are the tufted capuchins (*Sapajus* sp.) and squirrel (*Saimiri* sp.) monkeys [27,28,29,30], spending ~50% of their time in temporary but stable groupings [29]. Tufted capuchins (hereafter capuchins) are approximately three times the size (mean weight ~3 kg) of squirrel monkeys (mean weight ~0.74 kg), and as such, they are considered to be dominant over them [31]. Furthermore, capuchins are generally found to be tolerant towards subordinate others, including non-kin and other species [32,33], and compared to many cercopithecines such as rhesus macaques (*Macaca mulatta*) [34], and olive baboons (*Papio anubis*) [35], both capuchins and squirrel monkeys exhibit relatively low rates of aggression [30,36]. Perhaps it is for these reasons that this species combination has been successfully housed in captivity [37,38,39]. The two groups of Guianan tufted capuchins (*Sapajus apella*) and Guianan squirrel monkeys (*Saimiri sciureus*) housed at the Living Links Research Centre, Royal Zoological Society of Scotland at Edinburgh Zoo, UK (hereafter Living Links—see Figure 1) have been living together since 2008. While mixed-species exhibits often benefit from larger and more naturalistic enclosures, comparatively, the inhabitants are still forced into much closer proximity than they would be in the wild. Furthermore, with little to no risk of predation and food provisioning, more time can be allotted to intra- and perhaps inter-specific interactions. But without the ecological pressures that favour grouping in the wild, can merely living together in a shared space result in the formation of mixed-species groups?

Here, we report on the use of social network analysis as a method to quantify the ‘groupness’ of the two mixed-species groups at Living Links. Previous research has found that the monkeys display low rates of both inter-specific and intra-specific aggression, and affiliative behaviours such as play and grooming have rarely been observed [16,38,39], which is in keeping with observations in the wild [27,28,29,30]. Therefore, spatial proximity was considered as a proxy measure for network associations [40,41] within mixed-species networks, as individuals are typically unable to interact (e.g., cooperate) or react (e.g., flee from a predator) if they are not close enough (in proximity) to other individuals [8,42,43]. A review by Kasper and Voelkl [40] presented findings from a network analysis of 70 primate groups (30 different species; 36 captive; 28 wild), indicating that network density (Table 2) was generally high ($\overset{-}{x}$ = 0.57, range = 0.49–0.93). Accordingly, we predict that overall mixed-species network densities will be high (≥0.5). However, based on previous studies examining space use [44] and behavioural synchrony [45] in the Living Links monkeys, we expect there to be few associations between species within the mixed-species networks. As dominant individuals typically occupy more central roles in their networks [46,47,48], we predict that capuchins, as the larger and more dominant species, will demonstrate higher centrality measures than squirrel monkeys. Their size and tolerance toward other species make capuchins key connectors within the network. While previous research has focused on space use and behavioural synchrony to assess groupness in the Living Links monkeys [44,45], the present study employs social network analysis as a novel tool for quantifying ‘groupness’ in mixed-species groups. Social network analysis offers a more nuanced understanding of the social dynamics between species, complementing previous methods [44,45]. Therefore, we aim to evaluate the effectiveness of social network analysis in measuring ‘groupness’ for the first time in this context.

## 2. Materials and Methods

### 2.1. Study Subjects

Two groups of Guianan brown tufted capuchins (*Sapajus apella*) and Guianan squirrel monkeys (*Saimiri sciureus*) housed in a mixed exhibit at Living Links (Figure 1) were studied. The West group consisted of 18 capuchins and 9 squirrel monkeys, and the East group consisted of 17 capuchins and 17 squirrel monkeys (Table 1). Both West and East sides include separate indoor enclosures for capuchins (7 m × 4.5 m × 6 m high) and squirrel monkeys (5.5 m × 4.5 m × 6 m high—with size restricted entrance/exit so capuchins cannot enter), as well as a shared outdoor enclosure of approximately 900 m^2^. Indoor temperatures were approximately 24 °C, maintained using radiant ceiling heaters and an air conditioning system, while outdoor temperature ranged between 12 °C and 25 °C during the study period. For daily husbandry routines, see [38].

### 2.2. Observational Sampling

We recorded 183 h of focal observations (three hours per individual, not including instances when individuals were observed in proximity during focal follows of other individuals) between April and August 2015. Monkeys were sampled evenly between 09:00 and 17:30, using instantaneous point sampling methods following Martin and Bateson [49]. The focal animal was selected at random, and the observer (SD-S) would count 5 s before beginning recording to ensure that there was no choice or visibility bias in the observed grouping(s). Each focal was recorded at 1 minute intervals for 10 minutes and sampled 18 times over the study period. In each point sample, all group members (capuchins and squirrel monkeys) within two body lengths (~40 cm, based on the size of the largest monkey in the group) from the focal monkey were noted [50,51]. This measure was applied to both species to standardise the measure for the purpose of mixed-species measures. If no monkey was within two body lengths, then the focal was described as solitary. Two body lengths were used as a cut-off point for social proximity as this was often the minimum distance that an individual allowed another individual to approach before they reacted (e.g., attend to or move away; pers. obs.). If a focal was out of sight for more than three minutes, then the focal was abandoned and re-sampled at another time. For mixed-species group analysis, where affiliative or aggressive interactions between species may be rare, building proximity networks is appropriate. In order for behaviour(s) and social information to transfer between individuals, they typically need to be in proximity, which is why proximity is a commonly used measure in primate social network analysis [16,40,52].

### 2.3. Data Analysis

All statistical tests were performed using the software package R (v.4.4.1) in the RStudio environment ([53,54]—see code and data https://osf.io/kax6s/?view_only=1b5e1bd3600345a69cc82d37ecfa9e23, accessed on 19 November 2024), with an α of 0.05 [55] for the overall network analysis. For the multiple testing of specific network metrics, we applied a Bonferroni correction, adjusting the significance level to 0.0125 (0.05/4). The *p*-values from the permutation tests were calculated as the proportion of cases in the distribution from the randomised datasets that were equal to or more extreme than the observed measure. We created networks using the asnipe package [56], and the package igraph [57] to calculate network metrics.

Association data were collected by repeatedly sampling the Living Links groups, recording who was in proximity (associating) with whom in each sampling period. We calculated association rates between dyads, using a simple ratio index (SRI) as described by Hoppitt and Farine [9]:$$x/(y_{a}+y_{b}+y_{ab}+x),\;\mathrm{assuming}\;y_{Null}=0$$
where x is the number of sampling periods with a and b observed associated, $y_{a}$ is the number of sampling periods with just a identified, $y_{b}$ is the number of sampling periods with just b identified, $y_{ab}$ is the number of sampling periods with a and b identified but not associated and $y_{Null}$ the number of sampling periods with neither a nor b identified. Values closer to zero indicate low association rates, while values closer to one indicate high association rates (i.e., individuals were observed more frequently in proximity).

We used permutation tests, following the approach outlined by Farine [56], to assess the significance of observed associations within the mixed-species network, using 1000 iterations of the data. In addition, we conducted permutation t-tests, also based on 1000 iterations, to compare key network metrics—degree centrality, eigenvector centrality, betweenness centrality, and clustering coefficient (see also Table 2)—between capuchins and squirrel monkeys in both the West and East groups. Permutation tests are particularly valuable when working with smaller, unequally distributed samples, as they do not rely on the assumption of normality and are more robust than traditional non-parametric tests that are often based on ranked data. Finally, the difference from the original data is compared to the differences from the permuted data. If the original difference is greater or less than most of the differences from the random permutations, this suggests that the observed difference is unlikely to be due to random chance.

Specifically, a permutation *t*-test is a non-parametric method that compares two groups (such as species) to determine if there is a significant difference in a given variable. First, the difference in means for a given variable is calculated from the original data between the two species. Next, the original values for each individual are randomly shuffled to create new datasets where the assignment of species to data points is random for x number of iterations, and the difference between the two permuted groups is calculated for each shuffle. Finally, the difference from the original data is compared to the differences from the permuted data.

Permutation tests and network diagrams were visualised using the ggplot2 package [58] in RStudio. A node refers to an individual monkey and is displayed as a square (capuchin) or circle (squirrel monkey) in the network diagrams. An edge (i.e., a line) indicates the frequency that a node was in proximity with another node (Table 2).

## 3. Results

During focal observations, the West group squirrel monkeys were rarely observed in shared enclosures with the capuchins (outdoor enclosure = 18%, capuchin indoor enclosure = 1%). In contrast, members of both species in the East group were observed for over half of focal observations in shared enclosures (outdoors = 34%; capuchin indoor enclosure = 20%). The overall network density for the East group was calculated to be 0.733, indicating a highly interconnected network where most potential associations were observed. In comparison, the West group had a lower network density of 0.590, suggesting a less interconnected structure with fewer associations between individuals (Figure 2). Furthermore, the mean association rates between dyads from the overall mixed-species network data were 0.032 (SD ± 0.056) for the West group and 0.022 (SD ± 0.04) for the East group. When compared to 1000 randomly generated networks, the association rates in both the East and West groups were significantly lower (*p* < 0.0001), indicating that the observed associations occurred less frequently than would be expected by chance.

To further assess the efficiency of communication and potential interaction within the mixed-species networks, we calculated the average path length, which measures the average number of steps along the shortest paths for all possible pairs of nodes in the network. In the context of these mixed-species networks, a ‘step’ represents moving from one node to another through proximity-based connections, where each step reflects the spatial closeness or how many individuals (or spaces) one must ‘pass through’ within the network. The average path lengths for the East capuchins and squirrel monkeys were both 0.006 (SD ± capuchins = 0.0003, squirrel monkeys = 0.001). In contrast, the average path length for the West capuchins was 0.013 (SD ± 0.004), whereas that for squirrel monkeys was 0.0.016 (SD ± 0.008). These results indicate that the East group has a more efficient and cohesive mixed-species network compared to the West group, with more opportunities for communication and interaction between species, as reflected by their shorter average path lengths.

When examining the network metrics derived from the original data for capuchins and squirrel monkeys (Table 3), we performed a series of permutation-independent samples *t*-tests based on 1000 iterations (Figure 3 and Figure 4). For both the West (observed difference = 0.463) and East (observed difference = 0.387) groups, capuchins exhibited significantly higher eigenvector centrality compared to squirrel monkeys (*p* < 0.0001), indicating that the observed differences are unlikely to be due to chance. When examining clustering coefficients, we found no significant difference for species in the East group (observed difference = −0.001, *p* > 0.05). In contrast, in the West group, capuchins were significantly more tightly clustered in the mixed-species network compared to squirrel monkeys (observed difference = 0.115, *p* < 0.0001), indicating that this difference is highly unlikely to have occurred by random chance.

For betweenness centrality, the West capuchins exhibited significantly lower betweenness centrality compared to squirrel monkeys (observed difference = −11.364, *p* < 0.0001), indicating that within their mixed-species network, squirrel monkeys are likely to play a more bridging role. The opposite trend was found in the East group (observed difference = 4.855, *p* < 0.0001), suggesting that the capuchins may serve as an important bridge between species in the mixed network. Regarding degree centrality, the East capuchins exhibited significantly lower degree centrality (observed difference = −1.529, *p* < 0.0001) compared to squirrel monkeys, indicating that the East capuchins associate with fewer individuals than the squirrel monkeys in the mixed-species network. The opposite trend was found in the West group, with capuchins displaying significantly higher degree centrality (observed difference = 7.333) than squirrel monkeys (*p* < 0.0001).

## 4. Discussion

Network densities for both the West and East groups were high (≥0.5), which is consistent with previous research on primate social network analysis [40]. However, despite these high densities, there was a lack of mixed-species aggregation, with network diagrams showing distinctly separate clusters for both the West and East capuchins and squirrel monkeys (Figure 2). These findings are in keeping with previous studies investigating the groupness of the Living Links mixed-species groups using space [44] and behavioural synchrony [45], which found that the monkeys were using different areas of shared enclosures (squirrel monkeys on the periphery and capuchins more central zones) and were asynchronous in their behaviours. The strength of ties between species in the West group was likely weaker due to the low percentage of time that squirrel monkeys were observed in shared enclosures with the capuchins (outdoor: 18%, capuchin indoor: 1%). In contrast, members of both species in the East group were observed for over half of focal observations in shared enclosures (outdoors = 34%; capuchin indoor enclosure = 20%). Additionally, the geodesic distance was visibly larger in the West sociogram compared to the East (Figure 2). However, despite these increased opportunities for coming into proximity and the shorter geodesic distance between clusters in the East group, the sociograms still showed distinct species clusters, similar to those in the West (Figure 2), which could indicate that the smaller squirrel monkeys may be actively avoiding coming into proximity with the capuchins, reinforcing the notion that simply having shared space does not translate to forming a mixed-species group.

The analysis of average path length within the mixed-species networks highlighted differences in network efficiency between capuchins and squirrel monkeys between groups. In the East group, both capuchins and squirrel monkeys had similar and short average path lengths, indicating that overall, the mixed-species network is efficient (i.e., individuals can interact with one another with minimal distance and effort, promoting quicker and more effective exchanges of information, social interactions or simply coming into proximity with one another) and cohesive with ample opportunities for communication and association between species. In contrast, in the West group, capuchins had shorter average path lengths compared to squirrel monkeys, suggesting that within their mixed-species network, capuchins are more efficiently connected, potentially facilitating quicker and more direct associations (i.e., coming into proximity more frequently) within the group. These differences highlight the importance of between-species associations and their role in shaping network connectivity and social dynamics within mixed-species groups.

At the species level, eigenvector centrality was significantly higher in the capuchins compared to squirrel monkeys in both the West and East groups, suggesting that capuchins are more centrally positioned within the overall network. This finding supports our prediction and indicates that the capuchins may be more influential (i.e., they are well-connected with other highly connected individuals) in forming connections within the mixed-species network.

When examining the West group, capuchins exhibited a significantly higher clustering coefficient than squirrel monkeys within the mixed-species network. This difference suggests that capuchins form more tightly connected subgroups, while squirrel monkeys are less cohesive within their own social structure as well as the overall mixed-species network. The lack of a significant difference in clustering coefficients between the East capuchins and squirrel monkeys suggests that both species exhibit similar levels of local connectivity or cohesion within the mixed-species networks. This implies that, on average, each species forms similarly cohesive subgroups with comparable tendencies to form tightly connected clusters. However, when examining the sociogram (Figure 2), there are three distinct clusters (one for the squirrel monkeys and two for the capuchins); therefore, while clustering coefficients appear similar across species, they do not necessarily indicate integration across species within the network. Instead, they point to a situation where the two capuchin clusters and the single squirrel monkey cluster form cohesive but separate subgroups within the overall network. Thus, cohesion within species does not necessarily lead to cohesion between species. Notably, when further examining the East sociogram (Figure 2), one of the capuchin clusters is a smaller group (*n* = 5). These individuals are unrelated to the alpha female and have comparatively low eigenvector centrality scores (≤0.2) to the rest of the capuchins (>0.3), which indicates that they are low in the hierarchy [48].

Furthermore, they are on the periphery of the main capuchin cluster and also the squirrel monkey cluster. If we consider Hamilton’s selfish herd theory [59] and one of the reasons for forming mixed-species groups being reduced risk of predation [2], then it makes sense for lower-ranking capuchins who will also likely be positioned on the periphery of wild capuchin groups to be in proximity with squirrel monkeys when in a mixed-species group. Additionally, in relation to the detection of threats, capuchins have been found to benefit from the vigilance of squirrel monkeys [28]. Moreover, Hirsch’s [60] study on wild capuchins found that vigilance by subordinate individuals was attributed to predator detection as well as social monitoring of dominant conspecifics. This suggests that vigilance for social monitoring enables subordinate individuals to avoid agonistic interactions. Therefore, within a captive environment where there is minimal to no risk of predation, being in proximity with squirrel monkeys (who are generally more vigilant) could benefit lower-ranking capuchins by alerting them to potential threats such as aggression from more dominant capuchins.

The higher betweenness centrality for the East capuchins indicates their role as important bridges between species in the mixed-species network, despite having fewer direct connections compared to the squirrel monkeys that had significantly higher degree centrality. This intermediary position means that capuchins, while less directly connected, play a role in linking different parts of the social network. This difference highlights how degree centrality and betweenness centrality capture different roles within the mixed-species network. Higher degree centrality, as seen in the squirrel monkeys, reflects extensive social integration and frequent direct associations, whereas higher betweenness centrality, as seen in the capuchins, captures a role that bridges separate clusters and connects otherwise isolated parts of the network. However, we found the opposite in the West group, where the capuchins had significantly greater degree centrality, engaging with more individuals than squirrel monkeys. This may simply reflect the larger number of capuchins (*n* = 18) to squirrel monkeys (*n* = 9) in this group, although they had the same opportunities as the East group to enter shared enclosure spaces. During focal observations, however, squirrel monkeys were rarely observed in shared spaces or in proximity to capuchins. Moreover, although the West squirrel monkeys had lower degree centrality, they exhibited higher betweenness centrality, suggesting that they occupy the bridging role within this mixed-species network.

Overall, it is evident from the social network analyses that neither the West nor East groups behaved or interacted as a fully cohesive mixed-species group. However, the East group showed higher levels of association, greater network efficiency, and shorter geodesic distances. While they did not form a fully integrated network, these factors indicate that the East group was closer to functioning as a mixed-species group than the West group. In contrast, the West group exhibited more distinct species clusters and a larger geodesic distance, reflecting less proximity and integration between the species. Additionally, the low percentage of time the squirrel monkeys spent in shared enclosures with the capuchins further supports the idea of more limited association and weaker social ties within this group. This likely contributed to the lower cohesion observed in the West group’s mixed-species network. Nevertheless, these observations highlight that in the absence of ecological pressures faced in the wild (e.g., risk of predation, limited food availability), animals that would typically form mixed-species groups are unlikely to do so in a captive environment.

However, there could be other factors contributing to a lack of proximity between species. For instance, the group composition of the West monkeys (capuchins = 18, squirrel monkeys = 9) was markedly different to the East that had equal numbers (*n* = 17), and as squirrel monkeys are much smaller than the capuchins, the size difference plus this discrepancy in group size may have reduced the likelihood of the squirrel monkeys coming into proximity with the capuchins. Furthermore, the environment itself may have contributed to a lack of proximity between species. For instance, Daoudi et al. [44] observed that the Living Links monkeys used distinct areas of their outdoor enclosure reflecting natural ecological niches [27]. Since these data were collected, refurbishments to the outdoor enclosures have been implemented to promote greater space use by both species, and group sizes have changed. These changes may impact the mixed-species networks, highlighting the importance of long-term monitoring, as such changes can have a considerable impact on the social dynamics of the group. Therefore, continuous assessment can support both behavioural understanding and the management of zoo husbandry, guiding decisions for optimal enclosure design and group composition in mixed-species exhibits.

## 5. Conclusions

In conclusion, social network analysis has proven to be a valuable tool for quantifying associations within mixed-species groups, providing important insights into how capuchins and squirrel monkeys associate in a captive setting. While both species largely formed separate clusters, indicating limited integration, the findings suggest that other factors beyond ecological pressures—such as group composition, enclosure design and individual species’ behavioural tendencies—might be influencing these social dynamics. Ultimately, our results highlight that in the absence of ecological pressures, such as predation risk and resource competition, which typically drive the formation of mixed-species groups in the wild [2], structural and social factors specific to captive environments become key. For example, imbalances in group size or species-specific preferences for habitat use may limit opportunities for aggregation and integration [44]. Furthermore, individual behaviour, rank and differences in social structures could also contribute to the persistence of distinct species clusters, even when shared space is available [45].

These findings suggest that while mixed-species groups in captivity may not form mixed-species groups as cohesive as expected, the use of social network analysis provides a nuanced understanding of these associations, allowing researchers to identify factors that either promote or hinder social cohesion. This approach offers valuable insights that could support zoo husbandry and management decisions regarding enclosure design and group composition. Future research should, therefore, continue to explore how these various factors influence mixed-species groups in captivity, potentially using more advanced network approaches such as multi-layer networks [19,61] to capture the complexity of social relationships and interactions within groups.

## Figures and Tables

**Figure 1 animals-14-03360-f001:**
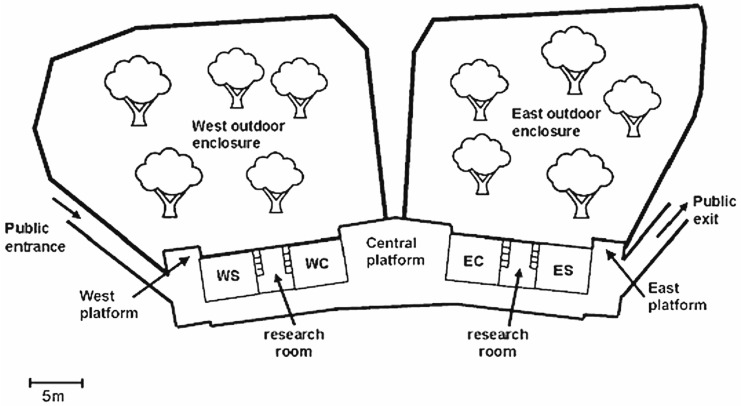
Schematic diagram (approximately to scale) of the Living Links West (WS = west squirrel monkey, WC = west capuchin) and East (ES = east squirrel monkey, EC = east capuchin) enclosures (adapted from [38]).

**Figure 2 animals-14-03360-f002:**
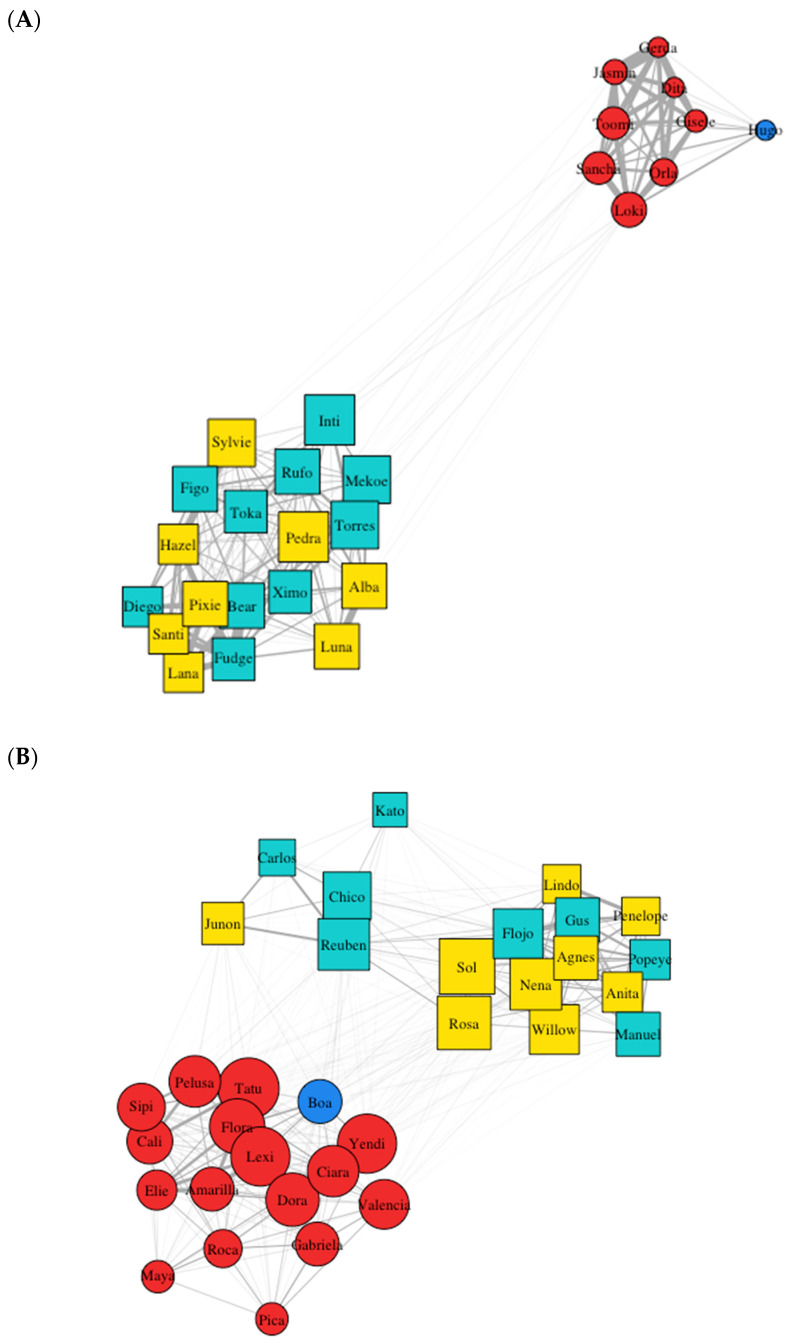
Mixed-species social networks for the (**A**) West and (**B**) East groups. Node shape is based on species: capuchins = square and squirrel monkeys = circle. Sex (capuchins: yellow = female; cyan = male; squirrel monkeys: red = female, blue = male). Node size is based on degree centrality, the strength of ties is based on the frequency of interactions between nodes and the distance between ties is based on the geodesic distance calculated as the sum of the weights of ties along the shortest path.

**Figure 3 animals-14-03360-f003:**
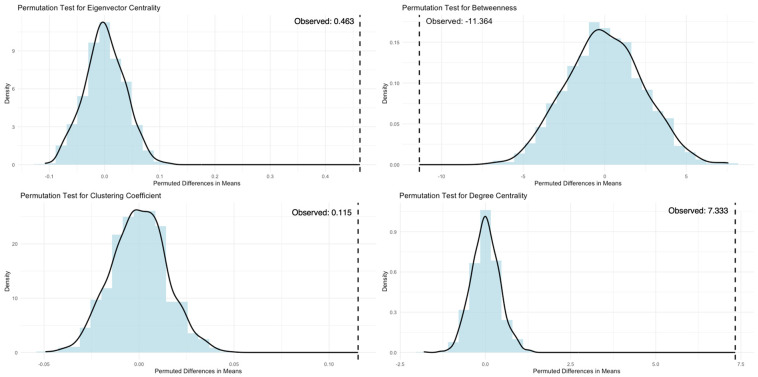
Permutation *t*-test distributions based on 1000 permutations of the network metrics (eigenvector centrality, clustering coefficient, betweenness centrality, and degree centrality) for the West capuchins and squirrel monkeys. Vertical dashed lines represent the observed difference.

**Figure 4 animals-14-03360-f004:**
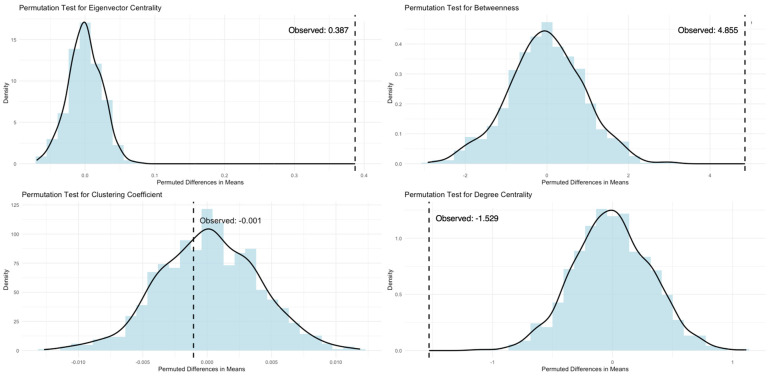
Permutation *t*-test distributions based on 1000 permutations of the network metrics (eigenvector centrality, clustering coefficient, betweenness centrality, and degree centrality) for the East capuchins and squirrel monkeys. Vertical dashed lines represent the observed difference.

**Table 1 animals-14-03360-t001:** Demographic information for the West and East capuchins (*Sapajus apella*) and squirrel monkeys (*Saimiri sciureus*) at Living Links.

Name	Species	Sex	Age Range (in Years) During Study	Exhibit
Diego	capuchin	M	12–13	West
Lana	capuchin	F	19	West
Santiago	capuchin	F	13	West
Sylvania	capuchin	F	11–12	West
Toka	capuchin	M	10	West
Figo	capuchin	M	9	West
Pedra	capuchin	F	7	West
Mekoe	capuchin	M	7	West
Inti	capuchin	M	5	West
Rufo	capuchin	M	5	West
Ximo	capuchin	M	5	West
Torres	capuchin	M	4	West
Luna	capuchin	F	4	West
Alba	capuchin	F	3	West
Mr Fudge	capuchin	M	1–2	West
Bear	capuchin	M	1–2	West
Hazel	capuchin	F	1	West
Pixie	capuchin	F	1	West
Hugo	squirrel monkey	M	4–5	West
Gerda	squirrel monkey	F	15	West
Jasmin	squirrel monkey	F	12	West
Toomi	squirrel monkey	F	8	West
Dita	squirrel monkey	F	5	West
Sancha	squirrel monkey	F	5	West
Orla	squirrel monkey	F	3	West
Gisele	squirrel monkey	F	2–3	West
Loki	squirrel monkey	F	1	West
Popeye	capuchin	M	13–14	East
Anita	capuchin	F	17	East
Junon	capuchin	F	14–15	East
Kato	capuchin	M	9–10	East
Manuel	capuchin	M	11	East
Penelope	capuchin	F	9	East
Carlos	capuchin	M	8–9	East
Chico	capuchin	M	6	East
Rosa	capuchin	F	5	East
Reuben	capuchin	M	4–5	East
Sol	capuchin	F	4–5	East
Flojo	capuchin	M	3–4	East
Lindo	capuchin	F	3–4	East
Willow	capuchin	F	2	East
Nena	capuchin	F	2	East
Gustavo	capuchin	M	1	East
Agnes	capuchin	F	1	East
Boa	squirrel monkey	M	9	East
Tatu	squirrel monkey	F	14	East
Roca	squirrel monkey	F	12	East
Maya	squirrel monkey	F	11	East
Elie	squirrel monkey	F	9	East
Cali	squirrel monkey	F	9	East
Pica	squirrel monkey	F	7	East
Yendi	squirrel monkey	F	6	East
Flora	squirrel monkey	F	6	East
Sipi	squirrel monkey	F	5	East
Lexi	squirrel monkey	F	4	East
Dora	squirrel monkey	F	4	East
Amarilla	squirrel monkey	F	3	East
Pelusa	squirrel monkey	F	3	East
Gabriela	squirrel monkey	F	3	East
Valencia	squirrel monkey	F	3	East
Ciara	squirrel monkey	F	3	East

**Table 2 animals-14-03360-t002:** Definitions of network metrics used in the analysis (adapted from [14]).

**Network Term**	**Description**
Node	Represents an individual within a network diagram (also referred to as vertex)
Tie	Relationship between two nodes in a network (also referred to as edge or arc)
Network density	The proportion of possible connections in a network that are actually present, measuring how interconnected individuals are within the network. A higher density means that more individuals are connected to each other, while a lower density indicates fewer connections.
**Association Indices**	**Description**
Degree Centrality	Measure based on the number of ties a node has. A higher degree of centrality means that the individual interacts frequently or maintains more consistent associations with other individuals in the network.
Eigenvector Centrality	Measure of the connectivity of an individual within its network according to the number and strength of connections and considering the centrality of the individuals it is connected to. Individuals with high eigenvector centrality are well connected to other individuals who are also well connected.
Betweenness Centrality	Measure of the number of shortest paths that pass through the considered individual (with the shortest path being the shortest distance, i.e., number of edges, between two nodes). These individuals serve as a bridge between others; therefore, an individual with high betweenness centrality connects different parts of the network, facilitating interactions across groups or clusters.
Clustering Coefficient	Measures the amount that a node tends to cluster with other nodes and captures the level of cohesion of the network. A high clustering coefficient indicates that an individual is embedded within a cohesive community, where their associates frequently interact or are associated with one another.
Average Path Length	The average number of steps (or connections) required to travel between all pairs of nodes (individuals) within the network. Shorter average path lengths indicate a more efficient and tightly connected network, while longer path lengths suggest less efficient connectivity.

**Table 3 animals-14-03360-t003:** Network metrics based on the original data for capuchins (West = 18; East = 17) and squirrel monkeys (West = 9; East = 17), mean (±SD).

Group and Species	Degree Centrality	Eigenvector Centrality	Betweenness Centrality	Clustering Coefficient
West capuchin	17.78 (1.31)	0.49 (0.33)	18.08 (1.31)	0.88 (0.09)
West squirrel monkey	10.44 (2.40)	0.02 (0.007)	29.44 (24.50)	0.77 (0.19)
East capuchin	23.41 (3.36)	0.48(0.31)	17.79 (9.34)	0.79 (0.03)
East squirrel monkey	24.94 (4.59)	0.09 (0.04)	12.93 (13.05)	0.79 (0.06)

## Data Availability

The data supporting the results reported in this study, along with the relevant code, are publicly available and can be accessed https://osf.io/kax6s/?view_only=1b5e1bd3600345a69cc82d37ecfa9e23, accessed on 19 November 2024.

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
