# Peer review of "Using Social Network Analysis to Assess ‘Groupness’ in a Mixed-Species Zoo Exhibit of Tufted Capuchins (Sapajus apella) and Squirrel Monkeys (Saimiri sciureus)"

_animals, 2024, doi:10.3390/ani14233360_

Round 1
Reviewer 1 Report
Comments and Suggestions for Authors
This MS describes using social network analysis (SNA) for mixed species exhibits of zoo species. Using SNA to assess the quality of a mixed species group is innovative, and could be of potential value. However, I wasn't sure exactly what that value was after reading through the MS. In the abstract, the authors discuss how measures of "groupness" are typically based on factors such as social interactions, group stability, proximity, etc, and propose that using SNA to assess association rates is an alternative method. But, they don't specifically compare "groupness" as assessed by SNA and the more traditional methods. In fact, SNA relied on proximity data, so it is unclear what the advantage is of using SNA instead of traditional methods. Further, it is not necessarily clear what having low association rates with heterospecifics means to the individuals in the mixed-species groups. Is their welfare lower compared to primates housed in single-species groups? Are the authors suggesting that the species should not be housed together if they show low association rates? I think the MS could be improved if the authors described the importance of interspecific association rates a bit more.
Other comments:
Introduction
Pg 2, line 51: it would be helpful if the authors could define group stability. They mention group stability as the first prerequisite for animal grouping but then later define it as the "complex social relationships and structure" (as the 4th prerequisite). It isn't clear why they list group stability twice, but it also isn't clear what they mean by group stability. Do they mean lack of aggression? Stable group membership?
Line 55-57: Can't patterns of association arise from a variety of factors other than shared hunger levels?
LIne 60-63: Many, if not most, species have a diverse repertoire of social behaviors- not just NHPs.
Line 82: It isn't clear why the authors included "affected by non-native species invasions" for fish, but didn't include this kind of modifier for bird or mammalian examples.
Line 87: Capuchins are considered to be dominant over squirrel monkeys when they are housed together, correct? Not just generally dominant?
Pg 3, line 115: The authors should explain what they mean by "network density". They might also explain other SNA terms such as centrality, etc.
Line 123- What is the importance of using SNA to measure "groupness"?
Materials and methods
Pg 3, Line 127: Above the authors talk about Guianan tufted capuchins. It would be ideal if they used the same term.
Pg 4: It is a bit hard to follow this section. When the observers noted a monkey close to the focal individual, did they note whether it was the same species? Was nearest neighbor based on the body length of the focal individual or the body length of the neighbor? Also, network density is not defined, but is described in the results, so it should be defined.
Line 156: What "major activities" were recorded? They are not detailed in this MS, so it is not clear why that is listed. If those behaviors were recorded for a different study, that should be made clear.
Line 162-164: I don't understand the explanation for p value. P values are not typically subject to different definitions across studies.
Line 168: Do calculations of association data take into account whether the neighbor was a conspecific or heterospecific?
Results: Figure 2: Squirrel monkeys are represented by circles, not triangles.
Discussion: Pg 8, line 184-287: What do the authors mean by network efficiency? What makes one association more efficient than another?
Pg 9, line 308: Does centrality = hierarchy status?
Author Response
Reviewer 1 comments and author responses
Initial comments: This MS describes using social network analysis (SNA) for mixed species exhibits of zoo species. Using SNA to assess the quality of a mixed species group is innovative, and could be of potential value. However, I wasn't sure exactly what that value was after reading through the MS. In the abstract, the authors discuss how measures of "groupness" are typically based on factors such as social interactions, group stability, proximity, etc, and propose that using SNA to assess association rates is an alternative method. But, they don't specifically compare "groupness" as assessed by SNA and the more traditional methods. In fact, SNA relied on proximity data, so it is unclear what the advantage is of using SNA instead of traditional methods. Further, it is not necessarily clear what having low association rates with heterospecifics means to the individuals in the mixed-species groups. Is their welfare lower compared to primates housed in single-species groups? Are the authors suggesting that the species should not be housed together if they show low association rates? I think the MS could be improved if the authors described the importance of interspecific association rates a bit more.
Response: Thank you for your thoughtful comments. We appreciate your recognition of the innovative approach of using social network analysis for assessing mixed-species exhibits. We have addressed your concerns and made revisions to improve the clarity and robustness of the manuscript. We have addressed comments and concerns from the other reviewers that are similar to some of yours, and thus the manuscript now reflects these changes.
Other comments:
Introduction
Comment: Pg 2, line 51: it would be helpful if the authors could define group stability. They mention group stability as the first prerequisite for animal grouping but then later define it as the "complex social relationships and structure" (as the 4th prerequisite). It isn't clear why they list group stability twice, but it also isn't clear what they mean by group stability. Do they mean lack of aggression? Stable group membership?
Response: Thank you for your comment, we acknowledge the inconsistency and need for clarity here. We have added the following text to the manuscript (see lines 50-61):
The first three require elaboration on how to define and measure whether these prerequisites are met. Group stability refers to the group maintaining a recognisable configuration over time, even as individual members join or leave. While the relative positions of individuals may shift, a degree of continuity among members is essential. The element of socialness may be directly observed by social attraction, where individuals seek out the presence of others in the group, which also serves to maintain the group. This association may be direct, such as physical contact or social behaviours such as play and grooming, or indirect, as through vocal communication, with physical proximity serving as a useful proxy for social engagement. Lastly, behavioural coordination in time and space implies that individuals are engaged in similar activities simultaneously and within a meaningful distance, allowing for both direct (e.g., physical) and indirect (e.g., vocal or visual) interactions.
Comment: Line 55-57: Can't patterns of association arise from a variety of factors other than shared hunger levels?
Response: Based on the literature cited and the theoretical background (i.e., Puliam’s double benefits of grouping) that states that ultimately two of the main reasons why animals group is for foraging advantages and reduced risk of predation, we feel that our sentence: “Patterns of association can arise due to individuals sharing identical or similar motivations and behaviours such as the degree to which they are hungry, thirsty or how they respond to a potential threat” is sufficient.
Comment: LIne 60-63: Many, if not most, species have a diverse repertoire of social behaviors- not just NHPs.
Response: We appreciate the reviewer’s point regarding the social complexity of various species. Our sentence uses the phrase 'socially complex species such as the non-human primates,' which is intended to highlight non-human primates as an example, rather than to exclude other species from being socially complex.
Comment: Line 82: It isn't clear why the authors included "affected by non-native species invasions" for fish, but didn't include this kind of modifier for bird or mammalian examples.
Response: Thank you for your comment, we have now made this clearer to provide more context (see lines 90-92): Thus far social network analysis has been applied to the study of mixed-species bird flocks [15,22-24], fish assemblages affected by non-native species invasions [25], and mixed-species groups of savannah herbivores [26].
Comment: Line 87: Capuchins are considered to be dominant over squirrel monkeys when they are housed together, correct? Not just generally dominant?
Response: Capuchins are indeed considered the more dominating species over the smaller squirrel monkeys. In the wild capuchins are known to prey on small mammals and birds. Capuchins will often displace squirrel monkeys and chase them away or show threatening behaviours such as lunging. Instances of aggression towards the squirrel monkeys is rare/infrequent.
Comment: Pg 3, line 115: The authors should explain what they mean by "network density". They might also explain other SNA terms such as centrality, etc.
Response: We have added a more detailed description of network terms in Table 2. Network density refers to the proportion of possible connections in a network that are actually present. In the context of social networks, it measures how interconnected individuals are within the network. A higher density means that more individuals are connected to each other, while a lower density indicates fewer connections.
Comment: Line 123- What is the importance of using SNA to measure "groupness"?
Response: We have added the following to the end of the introduction section, which hopefully provides more clarity:
While previous research has focused on space use and behavioural synchrony to assess groupness in the Living Links monkeys [44-45], the present study employs social network analysis as a novel tool for quantifying ‘groupness’ in mixed-species groups. Social network analysis offers a more nuanced understanding of the social dynamics between species complementing previous methods [44-45]. Therefore, we aim to evaluate the effectiveness of social network analysis in measuring ‘groupness’ for the first time in this context. (see lines 133-139)
Materials and methods
Comment: Pg 3, Line 127: Above the authors talk about Guianan tufted capuchins. It would be ideal if they used the same term.
Response: We believe the reviewer may be referring to the use of different terms for tufted capuchins and squirrel monkeys. We referred to tufted capuchins and squirrel monkeys using general terms (Sapajus sp. and Saimiri sp.), as these species are widely studied across South America. Historically, tufted capuchins were classified as Cebus apella (now Sapajus apella), and squirrel monkeys as Saimiri sciureus. However, under revised classifications for South American primate species, the specific animals housed at Living Links are Guianan tufted capuchins (Sapajus apella) and Guianan squirrel monkeys (Saimiri sciureus), which is why these terms are used in the Methods section.
Comment: Pg 4: It is a bit hard to follow this section. When the observers noted a monkey close to the focal individual, did they note whether it was the same species? Was nearest neighbor based on the body length of the focal individual or the body length of the neighbor? Also, network density is not defined, but is described in the results, so it should be defined.
Response: We recognise and have addressed the discrepancy with nearest neighbour and proximity measures. Please refer to the section on Observational sampling. Please also see Table 2 for description of network density.
Comment: Line 156: What "major activities" were recorded? They are not detailed in this MS, so it is not clear why that is listed. If those behaviors were recorded for a different study, that should be made clear.
Response: We agree that this sentence should be removed, as affiliative and aggressive interspecific interactions were not included in our analysis. Due to the limited number of affiliative and aggressive interactions observed between species, we chose to focus exclusively on proximity data, as this provided a more consistent basis for analysis. We have revised the manuscript to clarify this focus and have removed any references to data not analysed within this study.
Comment: Line 162-164: I don't understand the explanation for p value. P values are not typically subject to different definitions across studies.
Response: We provided this comment since permutation tests do not rely on any assumption about the underlying distribution of the data. Instead, the p-value is calculated by comparing the observed test statistic to a distribution generated by repeatedly randomising the data (i.e., permuting the labels or values) and recalculating the test statistic for each permutation. The p-value is then the proportion of permutations that yield a test statistic as extreme as or more extreme than the observed statistic. We hope that the distinction is now clear.
Comment: Line 168: Do calculations of association data take into account whether the neighbor was a conspecific or heterospecific?
Response: We have now provided clarification on proximity measures (see lines 163-174):
The focal animal was selected at random, and the observer (SD-S) would count 5 seconds before beginning recording to ensure that there was no choice or visibility bias in observed grouping(s). Each focal was recorded at 1-minute intervals for ten minutes and sampled 18 times over the study period. In each point sample, all group members (capuchins and squirrel monkeys) within two body lengths (~40cm, based on the size of the largest monkey in the group) from the focal monkey were noted [50-51]. This measure was applied to both species to standardise the measure for the purpose of mixed-species measures. If no monkey was within two body lengths, then the focal was described as solitary. Two body lengths were used as a cut-off point for social proximity as this was often the minimum distance that an individual allowed another individual to approach before they reacted (e.g. attend to or move away; pers. obs.). If a focal was out of sight for more than three minutes, then the focal was abandoned and re-sampled at another time. (see lines 163-174).
Comment: Results: Figure 2: Squirrel monkeys are represented by circles, not triangles.
Response: We have updated this.
Comment: Discussion: Pg 8, line 184-287: What do the authors mean by network efficiency? What makes one association more efficient than another?
Response: Thank you for bringing this to our attention as it highlights an important aspect of our analysis that could enhance the robustness of our conclusions. We acknowledge that our initial statement regarding the efficiency of the networks was more suggestive than definitive.
The average path length in each network provides a single summary statistic, resulting in too few data points for robust inferential testing. We have included more detailed descriptions in Table 2 to help readers interpret these findings more effectively.
Reviewer 2 also commented that more clarification was needed for what the authors mean by efficient network: network efficiency refers to how effectively individuals in the network can communicate or interact with one another. A shorter average path length indicates that individuals can reach each other more quickly through fewer connections, which is a hallmark of an efficient network.
In our analysis, a 'step' is defined as the movement from one individual (or node) to another through their proximity-based connections. Essentially, it reflects how spatially close two individuals are and how many other individuals or spaces must be traversed to facilitate communication or interaction.
Thus, when we say that a network is "efficient," we mean that individuals can interact with one another with minimal distance and effort, promoting quicker and more effective exchanges of information or social interactions. A network with a shorter average path length is considered more efficient because it enables faster access to connections among individuals, fostering robust social dynamics.
Comment: Pg 9, line 308: Does centrality = hierarchy status?
Response: We have now clarified this (see lines 354-357):
Notably, when further examining the East sociogram (Figure 2) one of the capuchin clusters is a smaller group (n = 5). These individuals are unrelated to the alpha female and have comparatively low eigenvector centrality scores (≤0.2) to the rest of the capuchins (>0.3), which indicate that they are low in the hierarchy [48].

Reviewer 2 Report
Comments and Suggestions for Authors
The manuscript describes an investigation of the social network structure of two groups of mixed-species primates exhibited at the Edinburgh Zoo. The groups comprise two species, namely Guianan capuchin monkeys and Guianan squirrel monkeys; the species naturally form mixed-species groups in nature. The study is comprehensive and the science robust. However there are two major issues with the manuscript as it is now.
The lesser of the problems with the manuscript is that it is written using a substantial amount of jargon and acronyms are overused throughout (the use of acronyms is unnecessary and would go a long way to improving the readability of the manuscript). As pointed out in the direct comments, much of the social network analysis is framed in a technical manner, which makes understanding and interpreting the findings challenging. That is not to say that the authors are incorrect or are using inappropriate terminology – they are not – but rather that the translation from technical and statistical significance to biological and real-world significance is lacking. The authors should reframe these aspects of the study in biological terms, especially in the context of the discussion section. The authors do provide interpretation of their findings in relation to the biology of the species themselves but in using the technical terminology it creates a disjointed discussion which is not intuitively digestible for the reader.
The other issue lies in the novelty of the study. The authors refer to two previous publications describing studies of the same subjects which describe the spatial association between the two species. Based on the previous studies, the groups do not show behavioural association and the findings of Daoudi et al 2017 in particular seem to mirror the findings presented here and suggest active avoidance of the capuchin monkeys by the squirrel monkeys. Thus, given that the authors already had a published manuscript which showed little spatial association between the species, it then raises the question of why a second investigation of the spatial association between the two species in the same groups is warranted. The authors should rather reframe their study with a different research question and possibly consider using a non-spatial metric to analyse.
For these reasons I am recommending that the manuscript be reconsidered following major revision.

Author Response
Comments: The manuscript describes an investigation of the social network structure of two groups of mixed-species primates exhibited at the Edinburgh Zoo. The groups comprise two species, namely Guianan capuchin monkeys and Guianan squirrel monkeys; the species naturally form mixed-species groups in nature. The study is comprehensive and the science robust. However there are two major issues with the manuscript as it is now.
The lesser of the problems with the manuscript is that it is written using a substantial amount of jargon and acronyms are overused throughout (the use of acronyms is unnecessary and would go a long way to improving the readability of the manuscript). As pointed out in the direct comments, much of the social network analysis is framed in a technical manner, which makes understanding and interpreting the findings challenging. That is not to say that the authors are incorrect or are using inappropriate terminology – they are not – but rather that the translation from technical and statistical significance to biological and real-world significance is lacking. The authors should reframe these aspects of the study in biological terms, especially in the context of the discussion section. The authors do provide interpretation of their findings in relation to the biology of the species themselves but in using the technical terminology it creates a disjointed discussion which is not intuitively digestible for the reader.
The other issue lies in the novelty of the study. The authors refer to two previous publications describing studies of the same subjects which describe the spatial association between the two species. Based on the previous studies, the groups do not show behavioural association and the findings of Daoudi et al 2017 in particular seem to mirror the findings presented here and suggest active avoidance of the capuchin monkeys by the squirrel monkeys. Thus, given that the authors already had a published manuscript which showed little spatial association between the species, it then raises the question of why a second investigation of the spatial association between the two species in the same groups is warranted. The authors should rather reframe their study with a different research question and possibly consider using a non-spatial metric to analyse.
For these reasons I am recommending that the manuscript be reconsidered following major revision.
Response: Thank you very much for your positive feedback on the study’s relevance and scientific rigor. We greatly appreciate your insights and constructive suggestions.
In relation to your comments on jargon and acronyms, we agree that reducing jargon and minimising the use of acronyms will enhance accessibility, particularly for readers who may not be as familiar with social network analysis. We have replaced acronyms with full terms.
Regarding your concerns relating to the novelty of the study. To clarify, this manuscript builds directly on findings from my prior work and forms a component of my broader PhD research on mixed-species grouping in tufted capuchins and squirrel monkeys. The Daoudi et al (2017) and Daoudi-Simison et al. (2023) findings stem from my PhD research on space use and behavioural synchrony in mixed-species groups. PhD dissertations in the UK are not considered official publications; thus, this study provides additional analysis and insights that are distinct from the earlier published work. We have addressed this in the introduction to make it clearer.
“While previous research has focused on space use and behavioural synchrony to assess groupness in the in the Living Links monkeys, the present study employs social network analysis as a novel tool for quantifying ‘groupness’ in mixed-species groups. Social network analysis offers a more nuanced understanding of the social dynamics between species complementing previous methods. Therefore, we aim to evaluate the effectiveness of social network analysis in measuring ‘groupness’ for the first time in this context.” (lines 124-130).
Comment: Title - The species names should be italicised.
Response: Thank you for spotting this. We have updated the title so that the species names are in italics.
Introduction
Comment: Lines 41-42 While it may certainly be easier to write a manuscript using acronyms, it does not necessarily facilitate the reading thereof. Avoid the use of acronyms unless they are for well-established terms or replace a particularly complex phrase, such as a method of analysis or a chemical structure. MSG is also already associated with monosodium glutamate and so it does not seem sensible to use an acronym with a pre-existing association.
Response: Your feedback regarding the readability of the manuscript was helpful. As stated above, we agree that reducing jargon and minimising the use of acronyms will enhance accessibility, particularly for readers who may not be as familiar with social network analysis. We have replaced acronyms with full terms.
Comment: Line 111 Again, is it necessary to have another acronym? Why not simply describe the social network under consideration and then refer to it as 'networks' thereafter.
Response: As stated above, we agree that reducing jargon and minimising the use of acronyms will enhance accessibility, particularly for readers who may not be as familiar with social network analysis. We have replaced acronyms with full terms.
Materials and methods
Comment: Line 156, What behaviours were recorded? An ethogram should be provided to allow readers to understand exactly what was recorded. What is the significance of mentioning that behaviour was scored as well? These data are not used in the analyses nor are they used to aid interpretation in the discussion section either.
Response: Thank you for highlighting this. We agree that references to behaviour should be removed, as affiliative and aggressive interactions were not included in our analysis. Due to the limited number of these interactions observed between species, we focused exclusively on proximity data, which provided a more consistent and reliable basis for our analysis. We have revised the manuscript to clarify this focus and have removed any references to data not analysed within this study.
Discussion
Comment: Lines 275-280 These results feel as though they are understated. It seems that the squirrel monkeys are actively avoiding the capuchin monkeys by remaining in those parts of the enclosure that are free of capuchin monkeys (assuming that all individuals have freedom to move between indoor and outdoor sections). Even the East group seems to spend a third of their time, at most, in the presence of the capuchins, suggesting a distinct preference to be away from them. Coupled with the previous findings, this suggests a spatial segregation that is taking place on multiple fronts (i.e. enclosure selection as well as locations within a given enclosure) which, even without the inclusion of the social network analysis, suggests that the degree of association between the two species is low.
Response: We have addressed this in more detail as suggested (see lines 307-318).
Comment: Lines 283-286 What is meant by "network efficiency"? Efficient in what sense? Does this mean that the associations between individuals are closer overall (reflecting a tighter degree of social cohesion within the group)? The sentence in lines 286-287 suggests that it instead indicates a more flexible degree of association between individuals whereby individuals are more socially mobile. The meaning is not clear because the interpretive meaning of 'efficiency' is not clear.
Response: Again, as you suggest, we have addressed this in more detail. Network efficiency refers to how effectively individuals in the network can communicate or interact with one another. A shorter average path length indicates that individuals can reach each other more quickly through fewer connections, which is a hallmark of an efficient network.
In our analysis, a 'step' is defined as the movement from one individual (or node) to another through their proximity-based connections. Essentially, it reflects how spatially close two individuals are and how many other individuals or spaces must be traversed to facilitate communication or interaction.
Thus, when we say that a network is "efficient," we mean that individuals can interact with one another with minimal distance and effort, promoting quicker and more effective exchanges of information or social interactions. A network with a shorter average path length is considered more efficient because it enables faster access to connections among individuals, fostering robust social dynamics.
We have included more details in Table 2, and updated the discussion for clarity (321-326).
Comment: Lines 292-293 What "overall network" is being referred to here? Is this the intra-specific network (i.e. squirrel monkeys only or capuchin monkeys only) or the mixed-species network? This should be clarified.
Response: To clarify, "overall network" refers to the mixed-species network. In this context, we were specifically referring to the mean values for the squirrel monkeys, which are higher than those for the capuchins within the mixed-species network. We have updated the manuscript to clarify this point.
Comment: Line 304 – Here and throughout, what does "more connected" refer to? Is this to say that individuals have more social connections or that the strength of these connections is greater? Is it both? This needs to be clarified and written in more understandable terms for readers who are not familiar with social network analyses.
Response: We have tried to clarify the terms associated with social network analysis and have updated Table 2 to address your comments. In social network analysis, the term “more connected” typically refers to an individual having a higher number of social links or interactions with others within the network. To clarify, in the context of degree centrality, "more connected," refers to the number of direct connections (or in the case of our data the frequency of coming into proximity) an individual has with other individuals in the network. A higher degree centrality means that the individual interacts (comes into proximity) frequently or maintains more consistent associations. For betweenness centrality, this relates to individuals that serve as a bridge between others, so an individual with high betweenness centrality, plays a central role in connecting different parts of the network, allowing them to facilitate associations across groups or clusters. For eigenvector centrality, more connected refers not only to the number of connections an individual has but also the importance of those connections. This measure implies that an individual is connected to others who are themselves well-connected, which may indicate a central role in the social structure.
Comment: Line 329 The word "drastically" here feels heavy-handed. The only difference in the group composition was the number of female squirrel monkeys in the East group (half that of the West group). This should be reworded to more accurately reflect the differences in group composition.
Response: We have removed this word.
Comment: Lines 332-335 So there was a previous study that already showed that the groups segregated spatially? Why was this not mentioned in the introduction to frame the investigation? This is particularly important given that spatial relationships were used as the metric to quantify the associations between individuals.
Response: Thank you for your thoughtful comment. You are correct that the previous studies by Daoudi et al. (2017) and Daoudi-Simison et al. (2020) demonstrated spatial segregation between the two species in terms of space use and behavioural synchrony.
We acknowledge that these studies provide an important context for the current investigation, and we have included additional information in the introduction to more clearly reference these findings (see lines 124-130).
Comment: Lines 337-342, It is not clear what is being communicated here. What is the significance of the changes implemented after data collection ended? Were there obvious changes in the social networks or space-use of the monkeys following the changes that are informative to the study aim? If not, why is this being mentioned?
Response: What we were aiming to convey is that, while our analysis is based on the data collected at the time of the study, there have been changes in group composition and enclosure design since that period. Previous research, such as Leonardi et al. (2010), demonstrates that changes in enclosure design can impact the behaviour of these monkey groups. Shifts in social composition, group size, and environmental factors could potentially influence the social networks of the monkeys. This highlights the importance of continued monitoring. As social networks are likely to evolve over time with changes in group structure, long-term monitoring is essential to assess how these factors might affect the networks. Such monitoring may also be crucial for husbandry practices in captive environments. We have made this clearer in the manuscript (see lines 414-424).
Tables and Figures
Comment: Table 1 – As a general comment, it would be helpful to frame the association indices in a biological context. The terms used are technical and thus are not intuitive. Given the scope of the journal, it would be helpful to find ways either to describe or define the concepts using biological terms. For example, while ‘node’ may be the correct term to use from a technical standpoint, a reader will be more likely to understand the term ‘individual’ when describing the network structure. The descriptions for Eigenvector Centrality and Betweenness Centrality are particularly challenging to understand as a reader who does not have a detailed understanding of social network analysis. It might also be helpful to include a simple diagram to illustrate what each index is measuring. For example, if Betweenness Centrality is a measure of the degree of separation between two individuals, use the words ‘degree of separation’ as this will be easier for readers to interpret. Also, if a term is chosen and defined (e.g. tie), use the chosen term throughout, don’t use synonyms (i.e. edge or arc). Select a single term and use that one consistently.
Response: We appreciate your perspective on making the terminology more accessible to readers less familiar with social network analysis. However, we feel that maintaining the technical terms is important for accuracy and consistency in the field. To further aid understanding, we have added some additional context to the descriptions to explain what connectedness means in this context of the association indices measures. This now refers to Table 2 in the manuscript.
While we will not alter the technical terminology, we have made efforts to ensure that the descriptions are clear and intuitive, and we trust that this will strike an appropriate balance between precision and accessibility.
Comment: Figure 2 – The caption for the figure mentions the nodes as squares and triangles whereas the images are of squares and circles. This should be corrected.
Response: Thank you for pointing out the discrepancy in figure 2. We appreciate your careful attention to detail. We have corrected this replacing the description of squirrel monkey nodes as circles.
Reviewer 3 Report
Comments and Suggestions for Authors
This study presents valuable information regarding a topic, multi-species groups, that is widely used in captive environment yet poorly researched. The introduction is well written and does a good job at introducing the topic and the relevance of the study. In my opinion, the discussion gives useful interpretations of the results. I do have several comments regarding the methods, in particular regarding the measurement of proximity and some of the metrics used. The references included are appropriate overall, although there are few more recent papers that use proximity networks in primates that could benefit the manuscript. Overall, I believe that the research presented in this manuscript is of interest for all primatologists that work with MSG.
Comments regarding the method section:
1) Regarding proximity:
Did the study use nearest neighbour or spatial proximity to measure association? Line 137 states the former (the closest individual), while other parts of the manuscript (like 151) imply that all individuals within a distance were recorded. To clarify, nearest neighbour would mean recording only the nearest individual, even if more individuals are close to the focal. For example, if individual A is within 10cms of the focal and individual B is within 30cms, a nearest neighbour measure would only record individual A. However, using spatial proximity, both A and B would be recorded. This is an important point that needs clarification. If authors have used nearest neighbour but have data of all individuals in spatial proximity, I would highly recommend using the latter for the analysis as, in my opinion, it offers a more complete picture of the spatial association of all the individuals in the group.
In addition to this clarification I believe that the citations used in line 142 regarding the use of nearest neighbour in primates are not the most appropriate. I would argue that Croft et al. (2011) is not a good citation for this point, as it is not an empirical study of primate social networks. They do mention networks based on nearest neighbour, but they use other mammal species as examples. Dufour et al (2011) mention physical contact and spatial proximity (50cms) with other individuals, but not nearest neighbour. Lehmann & Ross (2011) do not mention using neither spatial proximity nor nearest neighbour to create networks. Instead, they report using social interactions. Lastly, Zhang et al (2012) use spatial proximity (1m) from other individuals, but do not mention nearest neighbour.
There are quite a few more recent examples of use of spatial proximity networks (although none, that I am aware, use nearest neighbour) in primate research that I would recommend. The work by de Lima and Ferreira (2021) is particularly relevant, as it studies proximity networks in capuchins:
Funkhouser, J. A., Mayhew, J. A., & Mulcahy, J. B. (2018). Social network and dominance hierarchy analyses at Chimpanzee Sanctuary Northwest. PLOS ONE, 13(2), e0191898. https://doi.org/10.1371/journal.pone.0191898
Díaz, S., Murray, L., Roberts, S. G. B., & Rodway, P. (2020). Social Network Analysis of a Chimpanzee (Pan troglodytes) Group in Captivity Following the Integration of a New Adult Member. International Journal of Primatology, 41(5), 683–700. https://doi.org/10.1007/s10764-020-00177-0
de Lima, V. C. C., & Ferreira, R. G. (2021). Social network changes during the development of immature capuchin monkeys (Sapajus spp.). Primates, 62(5), 801–815. https://doi.org/10.1007/s10329-021-00918-6
2) Regarding the permutation t-tests
It is important for authors to provide the full statistics for the tests instead of only the p-values, including effect size. I am not proficient enough using the package asnipe to know if it offers effect size measures. If not, common non-parametric tests such as Mann-Whitney can provide a measure of effect size while being more powerful than permutation tests in non-normal samples. As I understand, not all authors agree regarding the problem of ranking data in non-parametric tests (Weber & Sawilosky, 2009) or the advantages of permutation tests when accounting for the non-independent nature of network data (Hart et al., 2022).
Additionally, since authors perform quite a few tests, p-values should be adjusted for multiple testing. This is unlikely to change the interpretation of the results, but it is appropriate, nonetheless.
Relevant citations regarding permutation tests in case they can be helpful:
Hart, J. D. A., Weiss, M. N., Brent, L. J. N., & Franks, D. W. (2022). Common permutation methods in animal social network analysis do not control for non-independence. Behavioral Ecology and Sociobiology, 76(11), 151. https://doi.org/10.1007/s00265-022-03254-x
Weber, M., & Sawilowsky, S. (2009). Comparative Power Of The Independent t, Permutation t, and Wilcoxon Tests. Journal of Modern Applied Statistical Methods, 8, 10–15. https://doi.org/10.56801/10.56801/v8.i.397
Other comments:
Lines 156-157: This line specifies that all behaviours from the focal and neighbours were recorded. I would recommend removing this sentence from the manuscript for clarity, as these behaviours do not seem to be analysed in the study. However, if the authors registered affiliative behaviours, they could analyse affiliation networks as well.
Line 228: It would be good to include SD as well as the mean for path length.
Line 236: If possible, it is better to give the exact p value instead of p > 0.05.
Line 259: As the authors mention further down the text, this is the result of the number of squirrel monkeys (n=9) vs capuchins (n=18) in the west group. The degree of squirrel monkeys in this group is as high as it can get (all squirrel monkeys have the highest possible degree) and, despise of this, it is always going to be significantly smaller than the degree of capuchins. This makes degree, in my opinion, a measure that is not appropriate for this analysis, as it is highly conditioned by the number of individuals in a group. It works well for the east group as there is an equal number of individuals for each species, but it does not work when comparing species in the west group. I would recommend considering using strength centrality (sometimes called weighted degree) instead of degree, as it is less impacted by group size and its distribution represents individual participation in social association (see, for example, Kasper & Voelkl, 2009).
Lines 260-266: I wonder if these interpretations would be better suited for the discussion instead of the results.
Line 277: If space use is important for the discussion, it should be included in the results.
Paragraph starting in line 307: As a suggestion for the authors: I believe that running cluster analysis could be useful for the manuscript, as it would offer additional quantitative data to describe clusters and evaluate if the groups are divided into smaller clusters. While clustering coefficient offers information regarding the density of ties for each node’s ego network, hierarchical cluster analysis would offer information regarding how the group is composed of distinct subgroups.
Author Response
Comment: This study presents valuable information regarding a topic, multi-species groups, that is widely used in captive environment yet poorly researched. The introduction is well written and does a good job at introducing the topic and the relevance of the study. In my opinion, the discussion gives useful interpretations of the results. I do have several comments regarding the methods, in particular regarding the measurement of proximity and some of the metrics used. The references included are appropriate overall, although there are few more recent papers that use proximity networks in primates that could benefit the manuscript. Overall, I believe that the research presented in this manuscript is of interest for all primatologists that work with MSG.
Response: Thank you for your encouraging feedback on our manuscript. We are pleased that you found the study’s focus on multi-species groups relevant and valuable, particularly given the limited research in this area despite its prevalence in captive environments. We appreciate your positive remarks regarding our introduction and discussion sections, as well as your suggestion to explore additional, recent references on proximity networks in primates. We will review recent literature and incorporate relevant studies to enhance our manuscript’s context and analysis.
We also appreciate your comments on the methods, specifically regarding proximity measurement and metric selection, and will address each of these points in detail to ensure our methods are as clear and consistent as possible.
Comments regarding the method section:
Comment: 1) Regarding proximity:
Did the study use nearest neighbour or spatial proximity to measure association? Line 137 states the former (the closest individual), while other parts of the manuscript (like 151) imply that all individuals within a distance were recorded. To clarify, nearest neighbour would mean recording only the nearest individual, even if more individuals are close to the focal. For example, if individual A is within 10cms of the focal and individual B is within 30cms, a nearest neighbour measure would only record individual A. However, using spatial proximity, both A and B would be recorded. This is an important point that needs clarification. If authors have used nearest neighbour but have data of all individuals in spatial proximity, I would highly recommend using the latter for the analysis as, in my opinion, it offers a more complete picture of the spatial association of all the individuals in the group.
In addition to this clarification I believe that the citations used in line 142 regarding the use of nearest neighbour in primates are not the most appropriate. I would argue that Croft et al. (2011) is not a good citation for this point, as it is not an empirical study of primate social networks. They do mention networks based on nearest neighbour, but they use other mammal species as examples. Dufour et al (2011) mention physical contact and spatial proximity (50cms) with other individuals, but not nearest neighbour. Lehmann & Ross (2011) do not mention using neither spatial proximity nor nearest neighbour to create networks. Instead, they report using social interactions. Lastly, Zhang et al (2012) use spatial proximity (1m) from other individuals, but do not mention nearest neighbour.
Response: Thank you for your feedback regarding the description of proximity measures. We agree that there are contradictions in the manuscript, and we appreciate the opportunity to clarify and ensure consistency in the sections mentioned and throughout the manuscript.
We used point sampling methods as described in Martin & Bateson (2007) to systematically record proximity data. Specifically, all observable group members within two body lengths of the focal animal were recorded at 1-minute intervals during 10-minute observation sessions. If no monkey was within two body lengths at a sample point, the focal individual was classified as “solitary.” Observations were rotated across all group members to ensure balanced coverage of the entire group. This corrected description aligns with the methodology used in Morton et al. (2015, 2021). We have revised the manuscript to incorporate this clarified description of the proximity measure (see lines 156-164).
We have included the following references, as they both used 2-body lengths as a measure of spatial proximity.
Morton, F. B.; Weiss, A.; Buchanan-Smith, H. M.; Lee, P. C. Capuchin monkeys with similar personalities have higher-quality relationships independent of age, sex, kinship and rank. Anim. Behav. 2015. 105, 163-171.
Morton, F. B.; Buchanan-Smith, H. M.; Brosnan, S. F.; Thierry, B.; Paukner, A.; Essler, J. L.; ... & Lee, P. C.. Studying animal innovation at the individual level: A ratings-based assessment in capuchin monkeys (Sapajus [Cebus] sp.). J. Comp. Psychol. 2021,135(2), 258.
Comment: There are quite a few more recent examples of use of spatial proximity networks (although none, that I am aware, use nearest neighbour) in primate research that I would recommend. The work by de Lima and Ferreira (2021) is particularly relevant, as it studies proximity networks in capuchins:
Funkhouser, J. A., Mayhew, J. A., & Mulcahy, J. B. (2018). Social network and dominance hierarchy analyses at Chimpanzee Sanctuary Northwest. PLOS ONE, 13(2), e0191898. https://doi.org/10.1371/journal.pone.0191898
Díaz, S., Murray, L., Roberts, S. G. B., & Rodway, P. (2020). Social Network Analysis of a Chimpanzee (Pan troglodytes) Group in Captivity Following the Integration of a New Adult Member. International Journal of Primatology, 41(5), 683–700. https://doi.org/10.1007/s10764-020-00177-0
de Lima, V. C. C., & Ferreira, R. G. (2021). Social network changes during the development of immature capuchin monkeys (Sapajus spp.). Primates, 62(5), 801–815. https://doi.org/10.1007/s10329-021-00918-6
Response: Thank you for the suggested references. We have included de Lima and Ferreira (2021) (see 52 in reference list).
Comment: 2) Regarding the permutation t-tests
It is important for authors to provide the full statistics for the tests instead of only the p-values, including effect size. I am not proficient enough using the package asnipe to know if it offers effect size measures. If not, common non-parametric tests such as Mann-Whitney can provide a measure of effect size while being more powerful than permutation tests in non-normal samples. As I understand, not all authors agree regarding the problem of ranking data in non-parametric tests (Weber & Sawilosky, 2009) or the advantages of permutation tests when accounting for the non-independent nature of network data (Hart et al., 2022).
Additionally, since authors perform quite a few tests, p-values should be adjusted for multiple testing. This is unlikely to change the interpretation of the results, but it is appropriate, nonetheless.
Response: We have applied a Bonferroni correction, adjusting our significance level from 0.05 to 0.0125 (0.05/4) to account for the four permutation tests per group.
Relevant citations regarding permutation tests in case they can be helpful:
Hart, J. D. A., Weiss, M. N., Brent, L. J. N., & Franks, D. W. (2022). Common permutation methods in animal social network analysis do not control for non-independence. Behavioral Ecology and Sociobiology, 76(11), 151. https://doi.org/10.1007/s00265-022-03254-x
Weber, M., & Sawilowsky, S. (2009). Comparative Power Of The Independent t, Permutation t, and Wilcoxon Tests. Journal of Modern Applied Statistical Methods, 8, 10–15. https://doi.org/10.56801/10.56801/v8.i.397
Response: Permutation t-tests are more appropriate for our data rather than the non-parametric Wilcoxon and Mann-Whitney U tests you suggest. Our association matrix indices are continuous variables, and our primary focus is on comparing group means rather than medians. Permutation tests allow for a direct comparison of means while also utilising the full scale of the data, which is essential for our analysis. Previous research has shown that permutation tests are effective for analysing complex ecological data, particularly when traditional assumptions may not apply e.g.:
Croft, D. P., Madden, J. R., Franks, D. W., & James, R. (2011). Hypothesis testing in animal social networks. Trends in Ecology & Evolution, 26, 502–507.
Legendre, P., & Legendre, L. (2012). Numerical Ecology. Elsevier;
Miller, E. T., Farine, D. R., & Trisos, C. H. (2017). Phylogenetic community structure metrics and null models: A review with new methods and software. Ecography, 40, 461–477.;
We also thank the reviewer for recommending the paper by Hart et al. (2022). After reviewing it, we found that it primarily focuses on nodal and dyadic regression methods, which analyse relationships between variables under specific assumptions about data structure. In contrast, permutation t-tests are designed to assess differences in group means and thus do not involve the same assumptions or analytical framework. Off the back of this paper we also reviewed the Farine & Carter (2022) paper:
Farine, D. R., & Carter, G. G. (2022). Permutation tests for hypothesis testing with animal social network data: Problems and potential solutions. Methods in Ecology and Evolution, 13(1), 144-156.
They suggest using double permutation tests as a potential solution to address elevated Type I and Type II error rates that can arise when testing null hypotheses with social network data for regression analyses and correlational data. Therefore, the critiques and insights provided in Hart et al. (2022) and Farine & Carter (2022) do not directly apply to our analysis as we were not performing regressions.
Regarding the risk of Type I errors due to multiple tests, each social network measure (eigenvector centrality, betweenness centrality, clustering coefficient, degree centrality) captures distinct aspects of network structure and individual roles within the network. Analysing these metrics separately provides a comprehensive understanding of the social dynamics between the two mixed-species groups. However, to mitigate the increased risk of Type I errors, as mentioned above we have applied a Bonferroni correction, adjusting our significance level from 0.05 to 0.0125 (0.05/4) to account for the four permutation tests per group.
Permutation tests do not inherently provide an effect size, furthermore, due to the type of test effect size does not offer additional meaningful insights for our analysis. Therefore, we have chosen not to include effect size in our results.
While we appreciate the reviewers’ suggestions, we believe that the permutation t-test remains the most suitable method for our analysis, given the nature of our data and our research objectives.
Upon re-evaluating the manuscript, we identified some discrepancies in Table 2 (formerly Table 1), which necessitated re-running our analyses for the network metrics. This re-analysis has led to some adjustments in the Results and Discussion sections. While these changes minimally affect the reviewers' comments, we believe they have strengthened the robustness of our findings and overall manuscript.
Other comments:
Comment: Lines 156-157: This line specifies that all behaviours from the focal and neighbours were recorded. I would recommend removing this sentence from the manuscript for clarity, as these behaviours do not seem to be analysed in the study. However, if the authors registered affiliative behaviours, they could analyse affiliation networks as well.
Response: Thank you for pointing this out. We agree that this sentence should be removed, as affiliative and aggressive interspecific interactions were not included in our analysis. Due to the limited number of affiliative and aggressive interactions observed between species, we chose to focus exclusively on proximity data, as this provided a more consistent basis for analysis. We have revised the manuscript to clarify this focus and have removed any references to data not analysed within this study.
Comment: Line 228: It would be good to include SD as well as the mean for path length.
Response: Agree. We have added this.
Comment: Line 236: If possible, it is better to give the exact p value instead of p > 0.05.
Response: While we agree that lumped p-value categories are not of great use, we have opted to report p > 0.05 to indicate the non-significance of the result. In our opinion a non-significant finding is just that, even if it approaches 0.05, and this notation is commonly used within the literature to denote findings that do not reach statistical significance.
Comment: Line 259: As the authors mention further down the text, this is the result of the number of squirrel monkeys (n=9) vs capuchins (n=18) in the west group. The degree of squirrel monkeys in this group is as high as it can get (all squirrel monkeys have the highest possible degree) and, despise of this, it is always going to be significantly smaller than the degree of capuchins. This makes degree, in my opinion, a measure that is not appropriate for this analysis, as it is highly conditioned by the number of individuals in a group. It works well for the east group as there is an equal number of individuals for each species, but it does not work when comparing species in the west group. I would recommend considering using strength centrality (sometimes called weighted degree) instead of degree, as it is less impacted by group size and its distribution represents individual participation in social association (see, for example, Kasper & Voelkl, 2009).
Response: Thank you for your comment regarding the use of degree centrality in the west group. Upon further review, we found that the code was separating the network measures for each species, rather than considering their network positions within the context of the overall mixed-species network. As a result of this change, degree centrality would provide meaningful insights in this context, for in the case of the west group, where the number of squirrel monkeys (n=9) is smaller than the number of capuchins (n=18), the degree centrality highlights the fact that the squirrel monkeys are not making the most of all possible interactions. We have amended the results in Table 2 to reflect these changes.
Comment: Lines 260-266: I wonder if these interpretations would be better suited for the discussion instead of the results.
Response: We feel that some interpretation is necessary for the results section, but agree that lines 264-266 could be moved to the discussion section. However, the updated analysis has meant that the findings have changed and this section is no longer relevant. Updates have been made throughout the manuscript to reflect the changes.
Comment: Line 277: If space use is important for the discussion, it should be included in the results.
Response: Thank you for your comment. This has been moved to the beginning of the results section as descriptive statistics.
Comment: Paragraph starting in line 307: As a suggestion for the authors: I believe that running cluster analysis could be useful for the manuscript, as it would offer additional quantitative data to describe clusters and evaluate if the groups are divided into smaller clusters. While clustering coefficient offers information regarding the density of ties for each node’s ego network, hierarchical cluster analysis would offer information regarding how the group is composed of distinct subgroups.
Response: We acknowledge your suggestion that hierarchical cluster analysis could provide valuable insights into the potential substructure of the groups, but we feel that it is not directly relevant to the central questions of our study. Our primary focus is on the overall social network structure and the patterns of association between individuals within the mixed-species network.
The results from the network measures, including the clustering coefficient, already provide us with a sufficient understanding of the density and connectivity within the network, particularly in terms of species-level associations. Specifically, the clustering coefficient highlights the density of ties within individual nodes' ego networks, which effectively captures the key aspects of social interactions in this context.
Introducing cluster analysis would likely shift the focus away from the central research question and could complicate the interpretation without adding significant new insights for this particular analysis. Therefore, we have chosen not to pursue this additional analysis at this stage.

Reviewer 4 Report
Comments and Suggestions for Authors
This is an interesting, well-written paper that describes the proximity patterns of two mixed-species groups of capuchin and squirrel monkeys using social network analysis. Analyses suggest that despite sharing physical space with one another the different species were not effectively interacting with one another, a result that replicates the authors’ findings from previous studies. The authors suggest that the release of ecological pressures (e.g., no predation pressure, no difficulty finding food) at least partially accounts for the lack of cross-species social networks. An appropriate quantity of data was collected and the analyses appear to have been conducted correctly. The authors claim that the data from this study would be valuable when assessing interactions in mixed-species groups in captivity (primarily in zoological collections). While social network analyses are likely to be valuable in such conditions, it is not particularly clear that the data from this study would be. Some of the descriptions of the findings are a little confusing; at times the networks are referred to as cohesive, while at other times they are implied to be separate. Additional explanation for the way the data are interpreted would be valuable.
Materials and Methods
Lines 126-129 It might be valuable to provide some information concerning the age classes of the subjects. Additionally, were both squirrel monkey groups unimale groups? If so, was there a specific reason for this?
Lines 156-157 Were any analyses performed on the “behaviour (major activities)” of the focal and neighbor?
Results
Figure 2 legend Squirrel monkeys are circles, not triangles as stated.
Lines 225-232 Is there a statistical test result that supports more efficient networks in capuchins compared to squirrel monkeys?
Figures 3 and 4 are difficult to understand. Perhaps additional explanation of the figures could be provided in the legends?
Discussion
Lines 283-290 Do the authors think that the differences in social organization between the capuchins (multimale-multifemale groups) and squirrel monkeys (unimale-multifemale groups) might partially account for these differences?
Lines 291-306 It may be slightly problematic to discuss differences that were not statistically significant.
Lines 307-310 The relationship between the separateness of the cluster of the 5 capuchins and their purported low rank in the hierarchy is not clear.
Line 319-322 Are squirrel monkeys typically vigilant about aggression among capuchins?
Author Response
Comment: This is an interesting, well-written paper that describes the proximity patterns of two mixed-species groups of capuchin and squirrel monkeys using social network analysis. Analyses suggest that despite sharing physical space with one another the different species were not effectively interacting with one another, a result that replicates the authors’ findings from previous studies. The authors suggest that the release of ecological pressures (e.g., no predation pressure, no difficulty finding food) at least partially accounts for the lack of cross-species social networks. An appropriate quantity of data was collected and the analyses appear to have been conducted correctly. The authors claim that the data from this study would be valuable when assessing interactions in mixed-species groups in captivity (primarily in zoological collections). While social network analyses are likely to be valuable in such conditions, it is not particularly clear that the data from this study would be. Some of the descriptions of the findings are a little confusing; at times the networks are referred to as cohesive, while at other times they are implied to be separate. Additional explanation for the way the data are interpreted would be valuable.
Response: Thank you for your thoughtful and constructive feedback on our manuscript. We appreciate your positive remarks regarding the relevance and clarity of our study, which examines the proximity patterns of mixed-species groups of capuchin and squirrel monkeys through social network analysis. We are glad to hear that you found our analyses to be appropriate and aligned with our previous findings.
We acknowledge your concern regarding the interpretation of our results, particularly the descriptions of the networks as both cohesive and separate. We agree that further clarification is essential, and we have revised the manuscript to provide a more consistent interpretation of the data.
Materials and Methods
Comment: Lines 126-129 It might be valuable to provide some information concerning the age classes of the subjects. Additionally, were both squirrel monkey groups unimale groups? If so, was there a specific reason for this?
Response: We have included a table in the revised manuscript that outlines the age and sex classes of the monkeys observed (see Table 1).
Regarding the structure of the squirrel monkey groups, both were indeed unimale groups. This configuration was determined by the zoo's management during the time of data collection, and the authors did not have any influence over this arrangement.
Comment: Lines 156-157: Were any analyses performed on the “behaviour (major activities)” of the focal and neighbor?
Response: Due to the limited number of interactions observed between species, we focused exclusively on proximity data, which provided a more consistent and reliable basis for our analysis. We have revised the manuscript to clarify this focus in line with your comments and have removed any references to data not analysed within this study.
Results
Comment: Figure 2 legend: Squirrel monkeys are circles, not triangles as stated.
Response: We have updated this.
Comment: Lines 225-232: Is there a statistical test result that supports more efficient networks in capuchins compared to squirrel monkeys?
Response: Thank you for bringing this to our attention as it highlights an important aspect of our analysis that could enhance the robustness of our conclusions. We acknowledge that our initial statement regarding the efficiency of the networks was more suggestive than definitive.
The average path length in each network provides a single summary statistic, resulting in too few data points for robust inferential testing. We have included more detailed descriptions in Table 2 to help readers interpret these findings more effectively.
Reviewer 2 also commented that more clarification was needed for what the authors mean by efficient network: network efficiency refers to how effectively individuals in the network can communicate or interact with one another. A shorter average path length indicates that individuals can reach each other more quickly through fewer connections, which is a hallmark of an efficient network.
In our analysis, a 'step' is defined as the movement from one individual (or node) to another through their proximity-based connections. Essentially, it reflects how spatially close two individuals are and how many other individuals or spaces must be traversed to facilitate communication or interaction.
Thus, when we say that a network is "efficient," we mean that individuals can interact with one another with minimal distance and effort, promoting quicker and more effective exchanges of information or social interactions. A network with a shorter average path length is considered more efficient because it enables faster access to connections among individuals, fostering robust social dynamics.
Comment: Figures 3 and 4 are difficult to understand. Perhaps additional explanation of the figures could be provided in the legends?
Response: We understand that clarity is essential for effective communication of our results, and we appreciate your suggestion to expand the figure legends. In the revised manuscript, we have provided more detailed explanations in the figure legends to enhance understanding and ensure that the information presented is clear and accessible to readers.
Discussion
Comment: Lines 283-290 Do the authors think that the differences in social organization between the capuchins (multimale-multifemale groups) and squirrel monkeys (unimale-multifemale groups) might partially account for these differences?
Response: While this is an interesting point, we believe that the differences in social organisation may not fully account for the observed differences in social dynamics. If that were the case, we would expect to see more consistent patterns across both the East and West groups.
We feel that the number of individuals in each group likely has a more significant impact on social interactions. For instance, the male in the West squirrel monkey group was not very well connected, even among the squirrel monkeys themselves, which could suggest that a less established or more anxious male might influence the group's social structure. In contrast, the male in the East group appeared to be better integrated into the social network.
Since the update to the analysis this section has changed.
Comment: Lines 291-306 It may be slightly problematic to discuss differences that were not statistically significant.
Response: We have updated this section for clarity as it is based on the permutation t-test.
Comment: Lines 307-310 The relationship between the separateness of the cluster of the 5 capuchins and their purported low rank in the hierarchy is not clear.
Response: Thank you for your comment regarding the relationship between the separateness of the cluster of five capuchins and their low rank in the hierarchy.
Eigenvector centrality has been positively correlated with rank (Wooddell et al., 2020 – ref 48). These five individuals are indeed comparatively lower in rank than the rest of the group. Their lack of relatedness to the alpha female may partially explain their lower social standing within the group.
We have tried to make this clearer (see lines 350-353).
Comment: Line 319-322 Are squirrel monkeys typically vigilant about aggression among capuchins?
Response: Your question regarding the vigilance of squirrel monkeys towards aggression among capuchins is very interesting. Squirrel monkeys are considered a hyper-vigilant species (to do with their small size and susceptibility to higher predation than capuchins), and as such they are highly alert to potential threats, including aggressive interactions. Given the more confined spaces typical in captivity, it is likely that squirrel monkeys would exhibit heightened vigilance towards any aggression from capuchins.
Within their own species, squirrel monkeys often respond to aggressive interactions by observing or participating in what is commonly referred to as an "aggression ball." This behaviour involves high-pitched vocalizations and physical interactions among individuals, which can create a chaotic environment before the interaction eventually disperses.
Round 2
Reviewer 2 Report
Comments and Suggestions for Authors
The authors have clearly made an effort to address the comments presented for the previous version of the manuscript. In doing so, they have made the manuscript markedly more comprehensible for those not familiar with social network analysis and thus more accessible to the audience of the journal. There are three minor issues that remain to be addressed, following which the manuscript will be ready for publication.
Page 1
Line 34 What type of social ties are being referred to here. Inter-specific or intra-specific? State it explicitly.
Page 13
Lines 412-422 While the authors have argued that this section should remain, it still feels disjointed from the study and unnecessary; it does not add much apart from the point that ongoing monitoring will elucidate the mechanisms behind social network changes because changes in the environment may influence social networks. This could be stated in a single sentence, rather than a relatively lengthy description of events beyond the temporal scope of the study and for which the impact on the groups is speculative. That is not to say that the authors are incorrect - a large body of evidence from captive and free-ranging animals supports the idea that changes to the physical environment and group composition is likely to alter group cohesion and social functioning. It just feels that this may either overstated and be whittled down to a single statement or require elaboration to make it a more impactful discussion point.
Page 13
Lines 447-450 This sentence appears to be a repeat of the last sentence and should be removed.
Author Response
General comment: The authors have clearly made an effort to address the comments presented for the previous version of the manuscript. In doing so, they have made the manuscript markedly more comprehensible for those not familiar with social network analysis and thus more accessible to the audience of the journal. There are three minor issues that remain to be addressed, following which the manuscript will be ready for publication.
Response: Thank you for recognising the efforts we made to address the comments from the previous review. We are pleased that the revisions have enhanced the manuscript's clarity, and we greatly appreciate your acknowledgment of this improvement. Your feedback has been instrumental in helping us refine the manuscript, and we are grateful for your constructive and encouraging comments.
Comment: Page 1, Line 34, What type of social ties are being referred to here. Inter-specific or intra-specific? State it explicitly.
Response: We have included the word interspecific.
Comment: Page 13, Lines 412-422, While the authors have argued that this section should remain, it still feels disjointed from the study and unnecessary; it does not add much apart from the point that ongoing monitoring will elucidate the mechanisms behind social network changes because changes in the environment may influence social networks. This could be stated in a single sentence, rather than a relatively lengthy description of events beyond the temporal scope of the study and for which the impact on the groups is speculative. That is not to say that the authors are incorrect - a large body of evidence from captive and free-ranging animals supports the idea that changes to the physical environment and group composition is likely to alter group cohesion and social functioning. It just feels that this may either overstated and be whittled down to a single statement or require elaboration to make it a more impactful discussion point.
Response: we have shortened the section, but retained the key points, see changed below.
Lines 412-422 (of the clean revised version, resubmitted 13/11/2024) state:
For instance, Daoudi et al. [44] observed that the Living Links monkeys used distinct areas of their outdoor enclosure reflecting natural ecological niches [27]. found that the Living Links monkeys occupied different areas of their shared outdoor enclosure that mimicked ecological niches they would typically be observed in the wild [27]. Capuchins favoured more central zones containing tall trees and squirrel monkeys favoured peripheral zones containing evergreen bushes (similar to habitat in forest understory). It is worth noting that Ssince these data were collected refurbishments to the outdoor enclosures have been implemented to promote greater space use by both species, and group sizes have changed. Indeed, Leonardi et al [38] previously observed a reduction in interspecific aggression in the Living Links monkeys post refurbishment. Additionally, the West squirrel monkey group has increased to 16 individuals, while the East group has decreased to 14. These changes to the enclosures and in group size may impact the mixed-species networks, highlighting the importance of long-term monitoring, as such changes can have a considerable impact on the social dynamics of the group. Therefore, continuous assessment can support both behavioural understanding and the management of zoo husbandry, guiding decisions for optimal enclosure design and group composition in mixed-species exhibits.
Comment: Page 13, Lines 447-450, This sentence appears to be a repeat of the last sentence and should be removed.
Response: This sentence has been removed (line 393-394, of the clean revised version, resubmitted 13/11/2024).
Reviewer 3 Report
Comments and Suggestions for Authors
I am satisfied with the changes made by the authors. They have solved all my questions regarding proximity and the current manuscript presents clear and replicable explanation for this variable. I also consider that their reasoning for selecting permutation t-tests has merits and, therefore, I do not need it is necessary to perform non-parametrical tests instead.
As a minor comment, I believe the lines 447-450 were supposed to be deleted from the manuscript.
Author Response
General comment: I am satisfied with the changes made by the authors. They have solved all my questions regarding proximity and the current manuscript presents clear and replicable explanation for this variable. I also consider that their reasoning for selecting permutation t-tests has merits and, therefore, I do not need it is necessary to perform non-parametrical tests instead.
Response: We sincerely thank you for your positive feedback on our revised manuscript. We are delighted to hear that the changes made have satisfactorily addressed your concerns.
Comment: As a minor comment, I believe the lines 447-450 were supposed to be deleted from the manuscript.
Response: We have deleted the following sentence: ‘the East group demonstrated more characteristics of a mixed-species group compared to the West group. Specifically,’ as this repeats what is stated later on at the end (line 393-394, of the clean revised version, resubmitted 13/11/2024). We feel that the rest of the sentence should stay as it highlights the key differences between groups based on the findings from the network analyses and sociograms.